# OpenReview forum: "The Structural Safety Generalization Problem"
_NeurIPS.cc/2024/Workshop/SafeGenAi — SafeGenAi Poster_

### Official Review · Reviewer_6KY2 · 2024-10-09
**Reviews for  paper titled "The Structural Safety Generalization Problem"**

**Rating:** 6
**Confidence:** 4

**Review:**

Novelty: The authors propose that AI models exhibit vulnerabilities not only in terms of semantic content but also structural variations in inputs, even when the semantic meaning is preserved. This structural vulnerability includes issues like single-turn vs. multi-turn input or single-image vs. multi-image attacks. The novelty lies in framing this problem and showing that structural attacks represent a distinct and unsolved class of AI vulnerability.
Reasons to Accept: combining insights from adversarial attacks on language models, vision models, and Go-playing systems, to illustrate how structural vulnerabilities manifest across different AI systems.
Reasons to Reject: core discussion is mainly about vulnerabilities in general AI models, such as LLMs and VLMs. However, the relevance of these findings to generative models LLMs, vision-language models, and diffusion models—as emphasized in certain workshops, including Safe Generative AI—remains underexplored. The connection between the discussed vulnerabilities and generative AI applications should have been better articulated

---

### Official Review · Reviewer_wVUC · 2024-10-10
**"Structural safety" is an interesting claim, needs more elaboration about what is included and not included**

**Rating:** 8
**Confidence:** 3

**Review:**

Pros:
- The paper introduces structural safety generalization, a novel perspective on AI adversarial robustness. By multi-turn, multi-image attacks, the paper opens new avenues for examining vulnerabilities in AI systems. Previous research is mostly focused on single-turn or single-modality vulnerabilities.
- The experiments are well-executed, systematically evaluating vulnerabilities in multi-turn and multi-modal attacks. The authors provide evidence of the gaps in current AI models’ ability to generalize safety measures across different input structures.

Cons or suggestions:
- The paper attempts to connect multiple lines of research, such as low-resource language vulnerabilities, cyclic attacks in superhuman Go AIs, and multimodal/multi-turn vulnerabilities, under the umbrella of structural safety – where semantically equivalent inputs are not being treated equally by the model. However, the paper's framing of the issue (safety across various inputs that mean the same thing but attack via different structures) might be less appropriate in the multimodal context when perturbations are used because these perturbations introduce elements of input variability that are not strictly structural but might be more similar to bijective attacks. In these cases, the model’s failure to handle inputs similarly might be more about adversarial robustness to these bijective tricks, rather than a failure to generalize over input structure alone. What is included or not included in the claim of "different structure"?
- The nature of attacks for multimodal language models should be presented in the main text (rather than in Appendix N.2).
- Both multi-turn and multi-modal jailbreaks are known issues in the field, and similar findings were published concurrently. However, the paper still introduces new datasets which are useful and novel.